# Multiphysics simulation of magnetoelectric micro core-shells for wireless cellular stimulation therapy via magnetic temporal interference

**Ram Prasadh Narayanan**[1]*, **Ali Khaleghi**[1,2], **Mladen Veletić**[1,2], **Ilangko Balasingham**[1,2]

1 Institute of Electronic Systems, Norwegian University of Science and Technology, Trondheim, Norway,
2 Intervention Center, Oslo University Hospital, Oslo, Norway

* ram.p.narayanan@ntnu.no

**Data Availability Statement:** All relevant data is included in the paper. The generated data and the code used for generating certain results is available

## Abstract

This paper presents an innovative approach to wireless cellular stimulation therapy through the design of a magnetoelectric (ME) microdevice. Traditional electrophysiological stimulation techniques for neural and deep brain stimulation face limitations due to their reliance on electronics, electrode arrays, or the complexity of magnetic induction. In contrast, the proposed ME microdevice offers a self-contained, controllable, battery-free, and electronics-free alternative, holding promise for targeted precise stimulation of biological cells and tissues. The designed microdevice integrates core shell ME materials with remote coils which applies magnetic temporal interference (MTI) signals, leading to the generation of a bipolar local electric stimulation current operating at low frequencies which is suitable for precise stimulation. The nonlinear property of the magnetostrictive core enables the demodulation of remotely applied high-frequency electromagnetic fields, resulting in a localized, tunable, and manipulatable electric potential on the piezoelectric shell surface. This potential, triggers electrical spikes in neural cells, facilitating stimulation. Rigorous computational simulations support this concept, highlighting a significantly high ME coupling factor generation of 550 V/m·Oe. The high ME coupling is primarily attributed to the operation of the device in its mechanical resonance modes. This achievement is the result of a carefully designed core shell structure operating at the MTI resonance frequencies, coupled with an optimal magnetic bias, and predetermined piezo shell thickness. These findings underscore the potential of the engineered ME core shell as a candidate for wireless and minimally invasive cellular stimulation therapy, characterized by high resolution and precision. These results open new avenues for injectable material structures capable of delivering effective cellular stimulation therapy, carrying implications across neuroscience medical devices, and regenerative medicine.

from Github at https://github.com/mladenveletic/neuronStim.

**Funding:** Yes;

**Competing interests:** The authors have declared that no competing interests exist.

## Introduction

Stimulation therapy is the targeted application of controlled signals to biological tissues, predominantly focusing on responsive cells, such as neurons [1] to elicit specific physiological responses, offering therapeutic avenues for myriad medical conditions. By engaging specific neural pathways and circuits, stimulation therapy can modulate, rejuvenate, or regulate neural activity, potentially mitigating symptoms linked to neurological disorders [2, 3]. The recent advances with (non) invasive wearables and medical implants have boosted the research in wireless and battery-free solutions [4] for biological stimulation, sensing, communication and wireless power transfer [5–9]. Wireless stimulation therapy, either through micro-coils [10] or external wearable devices, which emit alternating magnetic or electrical fields for stimulation [11–13]. In this, Transcranial Magnetic Stimulation (TMS) is an FDA approved non-invasive form of brain stimulation that uses alternating magnetic field, as either a single pulse mechanism or as repetitive pulse modulation. TMS plays an important role in elucidating the plasticity of neural networks in the brain for therapeutic applications including Parkinson's Disease [14, 15], dystonia, stroke, deep brain stimulation (DBS) [16] and other psychiatric and neurological conditions [17]. However, it is not without concerns–issues related to penetration depth and potential side effects, like headaches and nausea, persists [17, 18]. On the other hand, transcranial Direct Current Stimulation (tDCS) involves the application of weak pulsating currents (1–2 mA) across the cortex using electrodes placed on the scalp [19]. Evidence suggests the efficacy of tDCS for specific neurological disorders, but different regions in the brain react differently to various neuromodulation signals, thus requiring better theoretical models and custom pulse modulations [20]. Moreover, tDCS also has evidence of side effects such as burning sensation, headache etc [21, 22]. These existing stimulation techniques remains unsuitable precise stimulation requirements (e.g. deep brain stimulation), due to the inability to penetrate the ganglion region of the brain, thus offering relief for conditions including paralysis rehabilitation [18, 19, 23], and connection to brain machine interface (BMI) systems [24, 25]. With the advances in micro-fabrication and the use of micro-electrode arrays for neuro-modulation, precise interaction with the physiological medium and subsequent electrical stimulation has improved the efficiency of neural sensing and the threshold of neural stimulation/modulation [26]. Electrophysiological stimulation utilizes microelectrode elements or arrays for precision interaction with biology, applicable across diverse scales— from organs to individual cells [27]. It is important on two fronts: it not only facilitates targeted interventions but also yields valuable physiological insights, acting as a conduit between electrical activity and biological responses. This method, supported by extensive research and practical implementations, holds promise for applications such as diagnosis and neuromodulation [28]. Powering these micro-electrode arrays and external stimulators could be through conventional wired bio-electrodes [29, 30] to contemporary wireless solutions [31–34]. The wired paradigm employs bioelectrodes tethered to an external power source. Though effective, it can be invasive, penetrating through tissues and vascular systems, carry inherent risks— breakage, infections, tissue damage, and other long-term complications [35]. In contrast, wireless modalities, characterized by their non-invasive nature, dispatch electrical pulses through the skin to activate underlying nerves [36, 37], although these techniques also have challenges related to limited penetration depths and complications with the channel and external manipulation of the wireless signal [38].

Using of TMS or tDCS techniques, with or without micro-coils or micro-electrode arrays, a trade-off exists in these methods between the focal depth and the intensity of the stimulation [39, 40]. For target regions confined to a specific area, a temporal interference signalling method is required not only to modulate and increase the focality of the applied signal but also

to reduce unnecessary stimulation of non-target areas [41]. Magnetic Temporal Interference (MTI) is a concept intertwined with TMS and tDCS to improve the focality and precision of stimulation through amplitude-modulated interference signalling [42, 43]. Recent studies have shown that with a four-coil MTI system, the maximum achieved depth, focality and the coil electric field intensity were 1.6 cm, 5.1 cm$^2$, and, 523.93 V/m, at a sinusoidal oscillation frequency of 10 Hz (Table 1, in [41]). The intensity of the induced electrical field is proportional to the applied frequency and the current intensity therefore requiring higher current intensity for the coil operation (~1kA) [41]. An intuitive hypothesis to address the gaps and challenges in the existing techniques of neural stimulation, external modulation, and wireless applicators, is to bridge the existing wireless coil based external control, to semi-implantable micro-coils or particle clusters or arrays. The implant micro coils, though are effective and precise, have limitation on the power consumption [44].

Magnetoelectric (ME) composites have been exploited for their combined property of magnetostriction and piezoelectricity, exhibiting magnetoelectric Multiphysics behaviour. ME composites, including micro and nanoparticles, have been simulated [45, 46], synthesized [47] and reported for their high ME coupling [48], to be used in various applications including, wireless power transfer [7, 49, 50], radiating antennas [51–55], bio-stimulation [37, 56], magnetic sensing [51, 57, 58] and targeted drug delivery [59–61]. In this paper, we introduce the concept of ME microparticles mediated MTI for minimally invasive and precise stimulation. We hypothesize that our system allows the use of high-frequency magnetic fields to achieve a higher electric current density closer to the cell/tissue surface. The ME composites could be embedded into tissue layers or be present as an array in the inner layers below the scalp. The overall idea is to utilise high frequency non-ionising magnetic fields with less implementation complexity, and generate low frequency demodulated, we electric field pulses from the ME core-shells. In a recent work, a composite combination of Magnetoelectric layers with a rectifying electron transport layer made of ZnO, was shown [62]. The ME effect of the composite material with the rectifying layer allows for the self-rectification of the resultant electrical field intensity. The noise behaviour and the non-linear coefficient was studied for a wide frequency range (1–30 kHz) for the use of ME laminates as magnetic field sensors. The carrier frequency was optimized to improve the field sensitivity of the ME laminate [63].

In our hypothesis, we investigate the use of ME non-linear property, for the active demodulation of the temporal interference frequency component from the two applied high-frequency fields, as shown in Fig 1. We show that the material non-linearity could itself be used for frequency demodulation and to produce multiple eigenmodes at their resonant frequencies with varied deformation patterns on their surface, for precise stimulation. This results in the generation of specific electric potential patterns as hotspots of higher electric field intensity on their surface. Moreover, we show that by utilizing the nonlinear magnetostriction of the core, we

**Table 1. Material properties of MetGlas (magnetostrictive core) and AlN (piezo shell).**

| Material property | MetGlas (Magnetostrictive Core) | Aluminium Nitride (AlN-Piezoshell) [3] |
|---|---|---|
| Electrical Conductivity | 7.25E5 S/m | 1E-6 S/m |
| Relative Permittivity | 1 | 9 |
| Young's Modulus | 152GPa | – |
| Poisson's ratio | 0.22 | – |
| Density | 7900 Kg/m$^3$ | 3300 Kg/m$^3$ |
| Saturation Magnetization | 700282 A/m | – |
| Saturation Magnetostriction | 12 ppm | – |
| Initial magnetic susceptibility | 200 | |

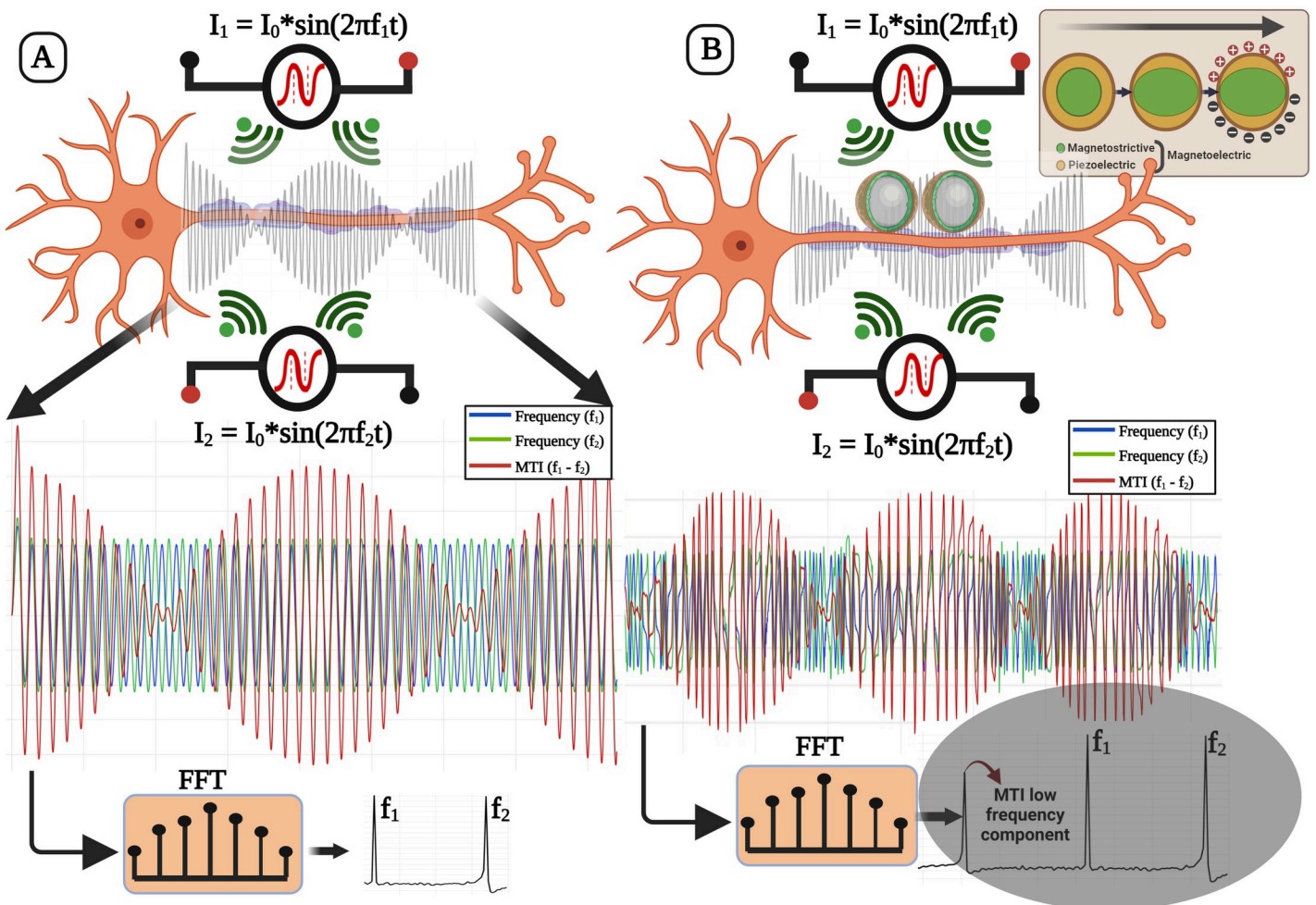

**Fig 1.** (**A. Schematic of conventional MTI**, where the power spectral density shows the presence of the two frequency components $f_1$ and $f_2$, and (**B., ME-MTI**, where the ME core-shells potentially function as a localized stimulant and a frequency demodulator due to their non-linear nature. This behavior of the (shown in the inset of B) demodulates the frequency component of the beating envelope, which could be utilized as a localized source for low frequency electrical stimulation. The power spectral density of the ME-MTI signal shows the demodulated MTI component (shown in the grey shaded area).

can demodulate amplitude-modulated higher frequency components that can directly interact with biological cells, enabling localized, low-frequency stimulation at the single-cell level. Moreover, these hotspots can be adjusted in intensity, facilitating high-resolution cell or tissue stimulation.

In this study, we examine the detailed design elements of size, geometry, and material properties, as well as the impact of magnetic fields on the intensity of the demodulated waveform for electrical stimulation. Additionally, the study focuses on interactions with biological tissues, particularly neurons, aiming for targeted and controlled stimulation.

The paper is organized into four sections. The materials and methods section begins with the analysis, explaining the materials and methods, including the simulation setup and governing equations for Multiphysics coupling. The results section presents results for magnetostriction, electrical field distribution, and the use of the ME microdevice for neural biostimulation. In the discussion section, we discuss the ME microdevice in comparison with existing literature. The last section concludes the paper and outlines future work.

## Materials and methods

Here, we discuss the materials and the composite equations used in COMSOL Multiphysics, to model the magnetostriction and the magnetoelectric behavior. This study primarily focuses on the simulation and modeling of an individual element of the ME composite. We have incorporated the non-linear behavior of the materials, especially when subjected to multi-frequency fields.

### Magnetoelectric physics

The property of magnetostriction, is the strain observed in ferromagnetic materials due to applied magnetic fields. The strain is a result of a strong orientation and weak perturbation of the electron spin in the ferromagnetic material, resulting in the lattice deformation of the ferromagnetic structures [24]. In other words, magnetostriction is caused due to the interaction of magnetic and elastic forces, thereby considered as a useful property for energy conversion [51]. Coupling with a piezoelectric domain will enable the subsequent conversion of the generated elastic forces into equivalent electrical fields. This Multiphysics is defined by the Joule and Villari effect which is expressed as [64],

$$\sigma = C\epsilon - e^T E - q^T H, \tag{1}$$

$$D = e\epsilon + kE + \alpha H, \tag{2}$$

$$B = q\epsilon + \alpha E + \mu H, \tag{3}$$

where, $C$ is the elastic stiffness tensor, $e$ and $q$ are the piezoelectric and piezomagnetic constant tensors respectively, $k$ and $\mu$ are the electric permittivity and the magnetic permeability tensors, $\alpha$ is the magnetoelectric coefficient tensor, $\epsilon$ and $\sigma$ are the mechanical strain and stress, respectively; $E$ and $D$, are the electrical field and displacement, respectively; and $H$ and $B$ are the magnetic field and flux density, respectively.

Multiphysics entities, piezoelectricity and magnetostriction are included with the coupling of solid mechanics, electrostatics, and magnetic fields modules. The magnetic field modeling was performed based on the constitutive B-H relation [64] wherein the boundary condition, *Ampere's Law*, was set for all the domains, except the magnetostrictive material as,

$$\vec{B} = \mu_0 \mu_r \vec{H}, \tag{4}$$

$$\nabla \cdot \vec{H} = \vec{J}. \tag{5}$$

The magnetostriction effect is defined with the boundary condition, *Ampere's Law, Magnetostrictive*, as,

$$\vec{B} = \mu_0 \left[ \vec{H} + \vec{M}\left(\vec{H}, S_{mech}\right) + \vec{M}_r \right], \tag{6}$$

where $B$ is the magnetic flux density, $H$ is the magnetic field intensity, $J$ is the volumetric current density, $\mu_0$ is the permeability of free space, $\mu_r$ is the magnetic relative permeability, $M$ is the magnetization, $S_{mech}$ is the stress tensor, and $M_r$ is the remanent magnetization, which is set as zero, since the material will not have residual magnetization after the external source is

removed. The magnetostrictive stress is modeled as a non-linear isotropic entity given as,

$$\epsilon_{me} = \frac{3}{2} \ \frac{\lambda_s}{M_s^2} dev(\vec{M} \otimes \vec{M}),$$ (7)

where, $\lambda_s$ and $M_s$ are the saturation magnetostriction and magnetization, respectively. The magnetization $M$, is given as a function of effective magnetic field intensity $H_{eff}$ and $M_s$ as,

$$\vec{M} = M_s L\left(|H_{eff}|\right) \frac{H_{eff}}{|H_{eff}|},$$ (8)

$$\vec{H}_{eff} = \vec{H} \ + \ \frac{3\lambda_s}{\lambda_0 M_S^2} S \vec{M},$$ (9)

where, $L$ is the Langevin function and $S$ is the stress tensor. For the piezoelectric shell, The *Charge Conservation* boundary condition governs the operating piezo shell given as,

$$\nabla \cdot \vec{F} = \rho_v,$$ (10)

where, $F$ is the electric flux density, and $\rho_v$ is the volume charge density.

## Simulation setup

To validate our initial concept of employing nonlinear magnetoelectric (ME) materials for biological stimulation, we have chosen a spherical core-shell geometry for our study. In this configuration, the spherical core is exposed to a magnetic field, resulting in deformations in all the directions that are fully coupled to the piezoelectric material comprising the shell. Using this geometry all the spherical vibration modes are transferred to the shell for maximum energy conversion. To validate, a single ME geometry is simulated to demonstrate the feasibility of generating a local voltage difference on the piezoelectric shell for potential cell and tissue stimulation.

Fig 2 depicts a schematic of the simulated COMSOL geometry, where two remote coils and the ME device are shown. The inset shows the magnetoelastic conversion of the induced magnetic field to mechanical stress and the subsequent piezoelectric coupling to produce an electric potential difference on the piezo shell. We used the commercially available 2628 MB MetGlas model as the magnetostrictive core due to its biocompatibility [45, 53] and high reported magneto mechanical coupling coefficient [65]. A piezoelectric coating of Aluminum Nitride (AlN) was added to the magnetostrictive core. The poling axis of the piezo shell was set to the same global axis coordinates. Table 1 provides the material characteristics of the selected magnetostrictive material (MetGlas) and piezoelectric material (ALN). Table 2 depicts the optimized dimensions of the microdevice. To model the piezoelectric material parameters, the material compliance ($s_E$), coupling matrix ($d$), and the permittivity ($\epsilon_T$) are required, and given as Table 3.

The contribution of losses is considered by replacing $\epsilon_T$ with $(1-jtan\delta)\,\epsilon_T$ and $c_E$ with $(1 + j\eta_s\,c_E$ in which $c_E = S_E^{-1}$. The loss values are given in Table 1.

We limited the dimensions of the simulated magnetostrictive core to 100 $\mu m$, making it suitable for use as an injectable device. In practical applications, this core could be incorporated into an array within a transplantable biological scaffold. The modeling methodology employed in our study is depicted in Fig 3. The core shell was placed between two symmetrically positioned coils driven by a constant AC current source of 100 $mA$. Two mechanical resonant frequencies (126 and 188 MHz) with a considerable frequency difference were selected as

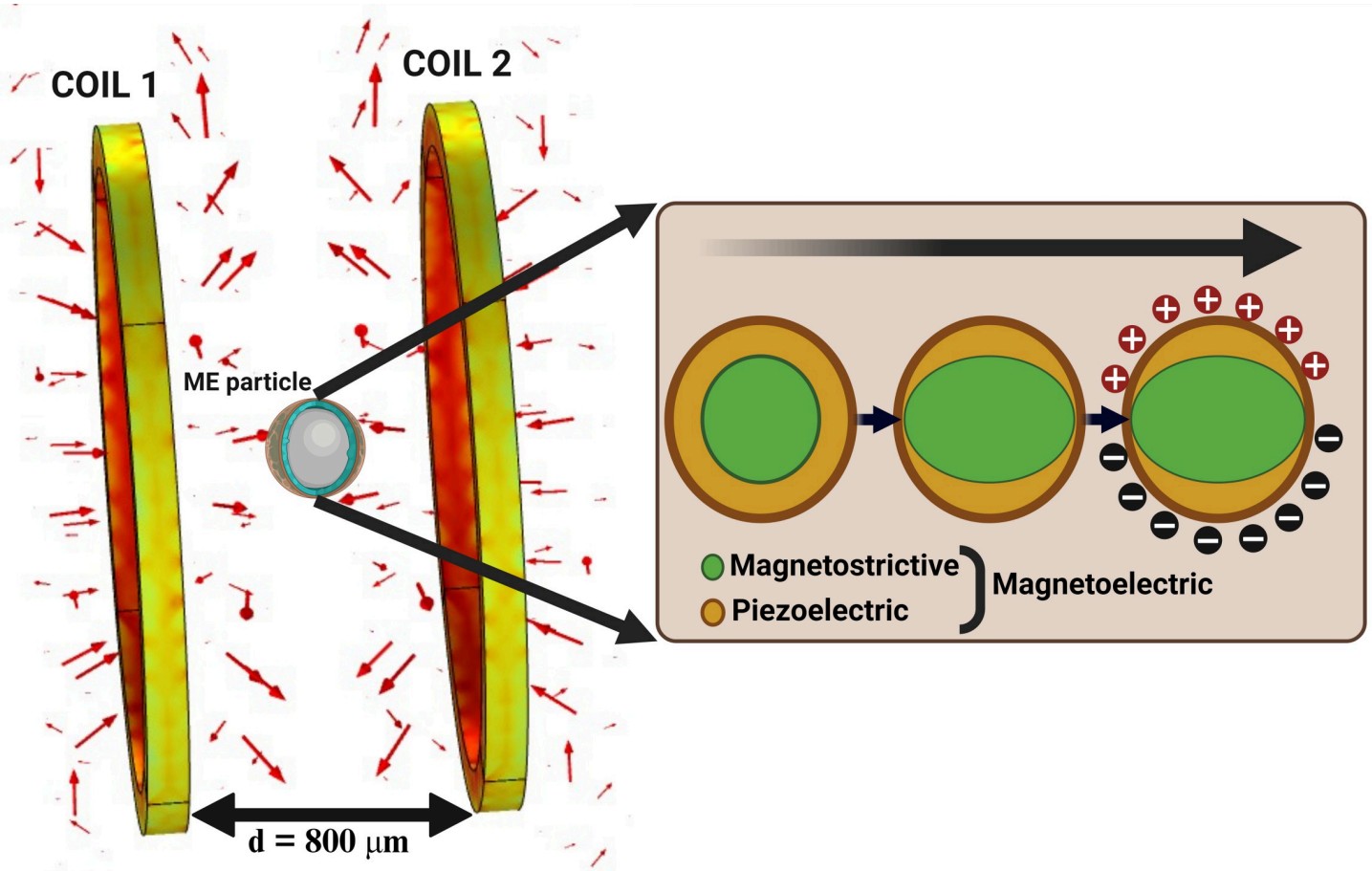

**Fig 2. Simulation setup of the COMSOL geometry showing the coils and the ME device.** The distance between the coils were set to 800 μm in the simulation. Both the coils are at an equal distance from each other and the ME particle. The MTI frequency, which is the difference frequencies of $f_1$ and $f_2$, from coils 1 and 2 respectively, is used to induce magnetostriction in the core, that results in the surface voltage.

the AC excitation of the magnetic fields applied to each coil. The choice of the AC excitation frequencies is detailed in the results section. We apply a user-controlled mesh, where triangular mesh is applied for the coil and particle domains, additionally, boundary layer meshing was added to the coil and the particle boundaries to increase the mesh density distribution. We were able to achieve an average mesh element quality of 0.8496.

Regarding the model simulation, the time step for the time domain analysis was set at $0.1 \times 0.05/(f_2)$, where $f_2$ is 188 MHz. Consequently, each time step (sampling time) is approximately in the order of 100s of nanoseconds in duration. This resulted in a higher number of

**Table 2. Design geometry of the particle.**

| Design geometry | Value |
| --- | --- |
| Magnetostrictive core diameter | 100 μm |
| Piezoelectric shell thickness | 37.5 μm |
| Magnetic field bias | 293 mT |
| Mechanical damping loss in the core and shell | 1E-4 |
| Dielectric loss in the piezoshell | 1E-4 |

**Table 3. Piezoelectric ALN material parameters: Material compliance ($s_E$), coupling matrix ($d$), and the permittivity ($\epsilon_T$).**

| Material compliance: $s_E \times 10^{-12}\left(\frac{1}{Pa}\right)$ | Coupling matrix: $d \times 10^{-12}\left(\frac{C}{N}\right)$ | Permittivity: $\epsilon_T$ |
|---|---|---|
| $\begin{bmatrix} 2.9 & -0.93 & -0.5 & 0 & 0 & 0 \\ -0.93 & 2.9 & -0.5 & 0 & 0 & 0 \\ -0.5 & -0.5 & 2.9 & 0 & 0 & 0 \\ 0 & 0 & 0 & 8 & 0 & 0 \\ 0 & 0 & 0 & 0 & 8 & 0 \\ 0 & 0 & 0 & 0 & 0 & 7.7 \end{bmatrix}$ | $\begin{bmatrix} 0 & 0 & 0 & 0 & -3.8 & 0 \\ 0 & 0 & 0 & -3.8 & 0 & 0 \\ -1.9 & -1.9 & 5 & 0 & 0 & 0 \end{bmatrix}$ | $\begin{bmatrix} 9 & 0 & 0 \\ 0 & 9 & 0 \\ 0 & 0 & 9 \end{bmatrix}$ |

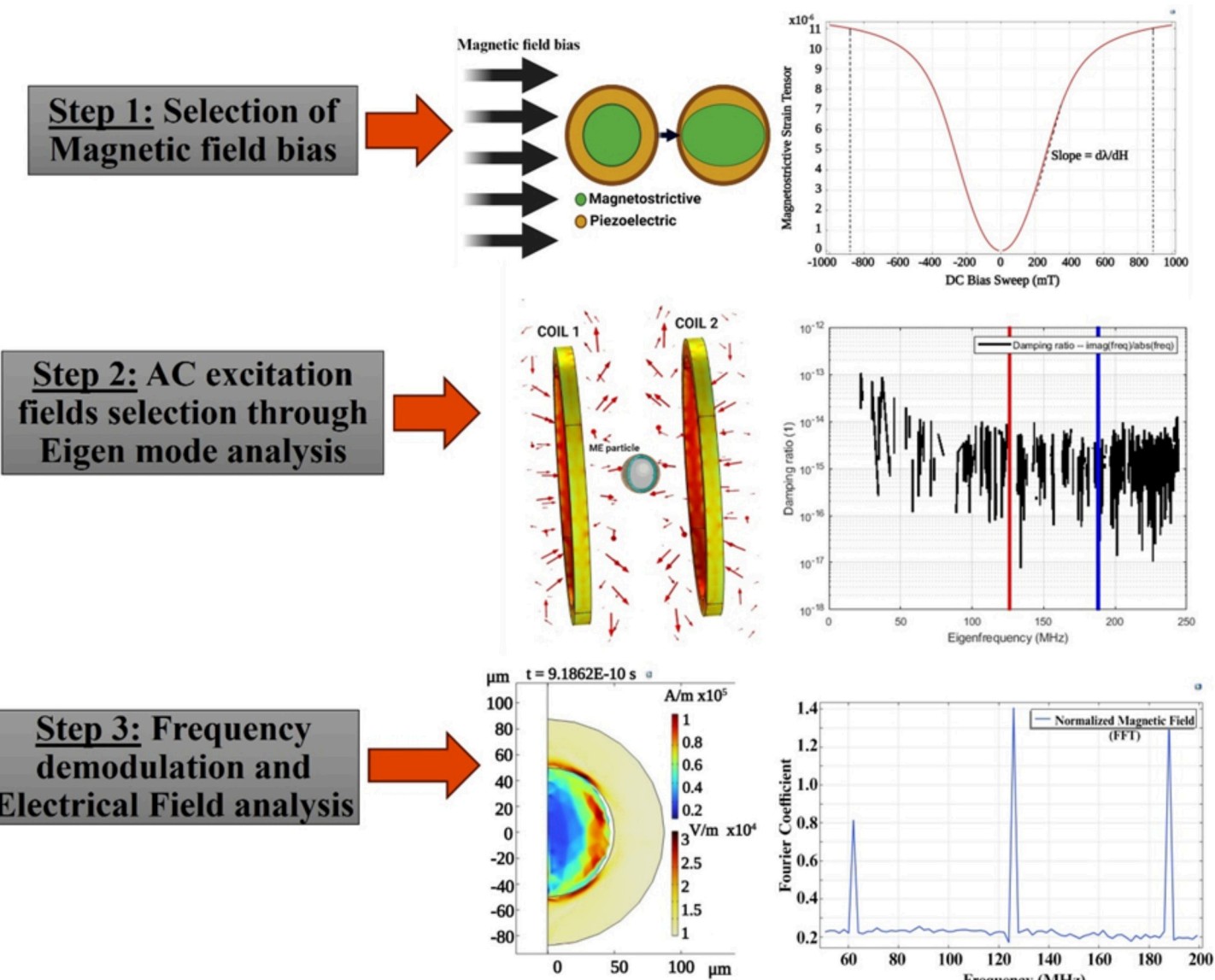

**Fig 3. Modeling method to analyze the non-linear frequency demodulation and the magnetoelectric coupling.** Step 1 shows the non-linear variation of Magnetostriction for an applied DC bias sweep. The DC bias field where maximum slope of Magnetostriction is observed, is selected as the Optimum DC bias, $H_{DC(opt)}$. Step 2 shows the selection of AC excitation field through Eigen mode analysis. For the simulation, we selected 126 and 188 MHz (shown by the red and blue vertical lines). In step 3, we analyze the frequency demodulation at the $H_{DC(opt)}$, and the selected AC excitations.

simulation steps, amounting to 200,000 entries, in the model. The number of degrees of freedom, with user-controlled meshing, was 7700 entries (refer to [66]), requiring around 24 GB (200000×7700×8B×2) of memory. Since this is a time-domain simulation, the factor of two is also multiplied because the time derivatives are also stored. Reduced sampling time would result in a decreased number of output times, which would subsequently reduce disk usage, albeit at the expense of diminished resolution of the time steps. The complexity of the ME Multiphysics and the requirement for time derivatives due to dual-frequency excitation are the primary reasons for the lengthy run times and high disk space consumption.

In contrast to the model presented in [45], which explores a magnetoelectric (ME) 'nanoparticle' using a frequency domain solver in COMSOL at a low frequency of 50 Hz, our design approach diverges significantly in multiple key aspects. First, we utilize micro-sized particles instead of nanoparticles. Second, our model operates at a very high frequency (VHF) range, which results in multi-modal magnetostrictive deformations—essential for creating high-field intensity on the particle. Additionally, we account for the nonlinear behavior of the magnetostrictive core, particularly when two frequencies are applied simultaneously to demonstrate MTI effect.

Given this complexity, our numerical methodology relies on a finite element time domain solver, specifically employing the highly nonlinear PARADISO solver settings in COMSOL. To improve computational efficiency, and demonstrate the nonlinear simulation features we selected two AC frequencies with a significant difference of 62 MHz, thus reducing the simulation's time steps and enabling a thorough demonstration of the ME core-shell's nonlinear functionalities. It is important to note that, in practical applications, the difference frequency should fall within the range suitable for biological stimulation response, specifically units of 100s of Hz (Table 1 in [60]).

## Results

We have organized our results into three distinct sections. First, we present findings related to the static magnetic field bias applied to our core-shell geometry. Second, we delve into the behavior of the magnetostrictive core when subjected to high-frequency AC field, focusing on frequency multiplication effects resulting from nonlinearity. Finally, we discuss the integration of the piezoelectric shell with the magnetostrictive core, which leads to electric field polarization and the conversion of mechanical deformations into electrical signals capable of interacting with biological systems.

### Static magnetic bias

The magnetostrictive core's elasticity-induced deformation, magnetostriction ($\lambda$) and subsequent electric polarization (on the piezo shell) are maximum when the ME device is operated at the optimum static magnetic bias $H_{DC(opt)}$. At this static magnetic bias, maximum magnetostrictivity $\left(d_\lambda = \frac{d\lambda}{dH_{DC(opt)}}\right)$ is also observed. MetGlas has a magnetostriction of 12 ppm and a saturation magnetization of 0.88T (700282 A/m) [67], and the magnetostrictive strain is limited to {-0.88, 0.88} T. Fig 4 shows the non-linear variation of the magnetostrictive strain tensor for applied static magnetic field in the range {-1, 1} T, for the selected MetGlas material (Table 1) and dimensions (Table 2). The optimum bias field $H_{DC(opt)}$ was identified in the range 200–350 mT (maximum sloop of the curve). Through the subsequent simulations, the optimum magnetic DC bias for producing maximum coupling between the piezo shell and the core was observed at 293 mT.

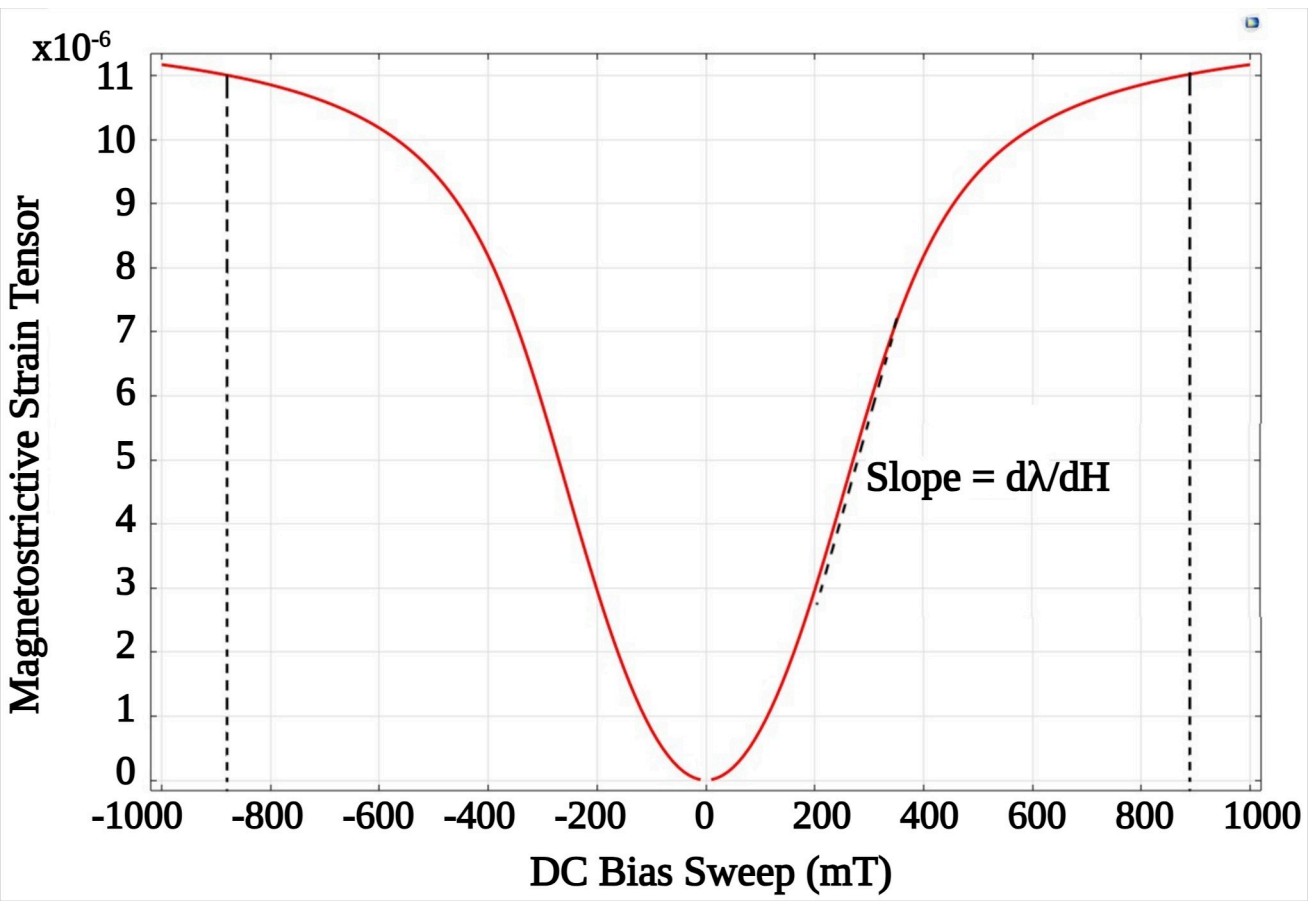

**Fig 4. Non-linear magnetization curve showing the rise and fall magnetostrictive strain at ≈ {-0.88, 0.88} T and the maximum slope of the magnetostriction in the range 200–350 mT.**

## AC excitation

At the mechanical resonant frequency, the magnetostriction, magnetization, and deformation of the magnetostrictive core are maximum [68]. Thus, the non-linear characteristic of the magnetostrictive device operating in its Eigen mode can produce irregular but intense deformation (in the core) and thus couple electrical hotspots on the subsequent piezo shell. This theory is directly related to the mechanical resonant modes' relationship with the elasticity caused by magnetostriction. Here, we analyzed the magnetostrictive sphere in its eigenmode mechanical resonances. In the simulation, the number of desired Eigen frequencies was set to 1000 and searched in the range of 30–240 MHz. The core shell geometry is biased at $H_{DC(opt)}$ of 293 $mT$. The damping ratio for the resulting Eigen frequencies was calculated as $\zeta = \frac{imag(freq)}{abs(freq)}$, as depicted in Fig 5.

This analysis was conducted to identify two resonant mode frequencies at which the damping ratio ($\zeta$) is extremely low ($<10^{-10}$) (i.e., the device is nearly "critically damped"), thus providing maximum mechanical deformation. As displayed in Fig 5, multiple combinations of such frequencies exist. We opted for two resonant frequencies with a significant difference, specifically 126 MHz and 188 MHz, to enable faster simulations and reduce computation time. Selecting frequencies closer to each other would necessitate smaller timesteps and extended simulation time. Given that the focus of this work is to provide simulation proof for non-linear

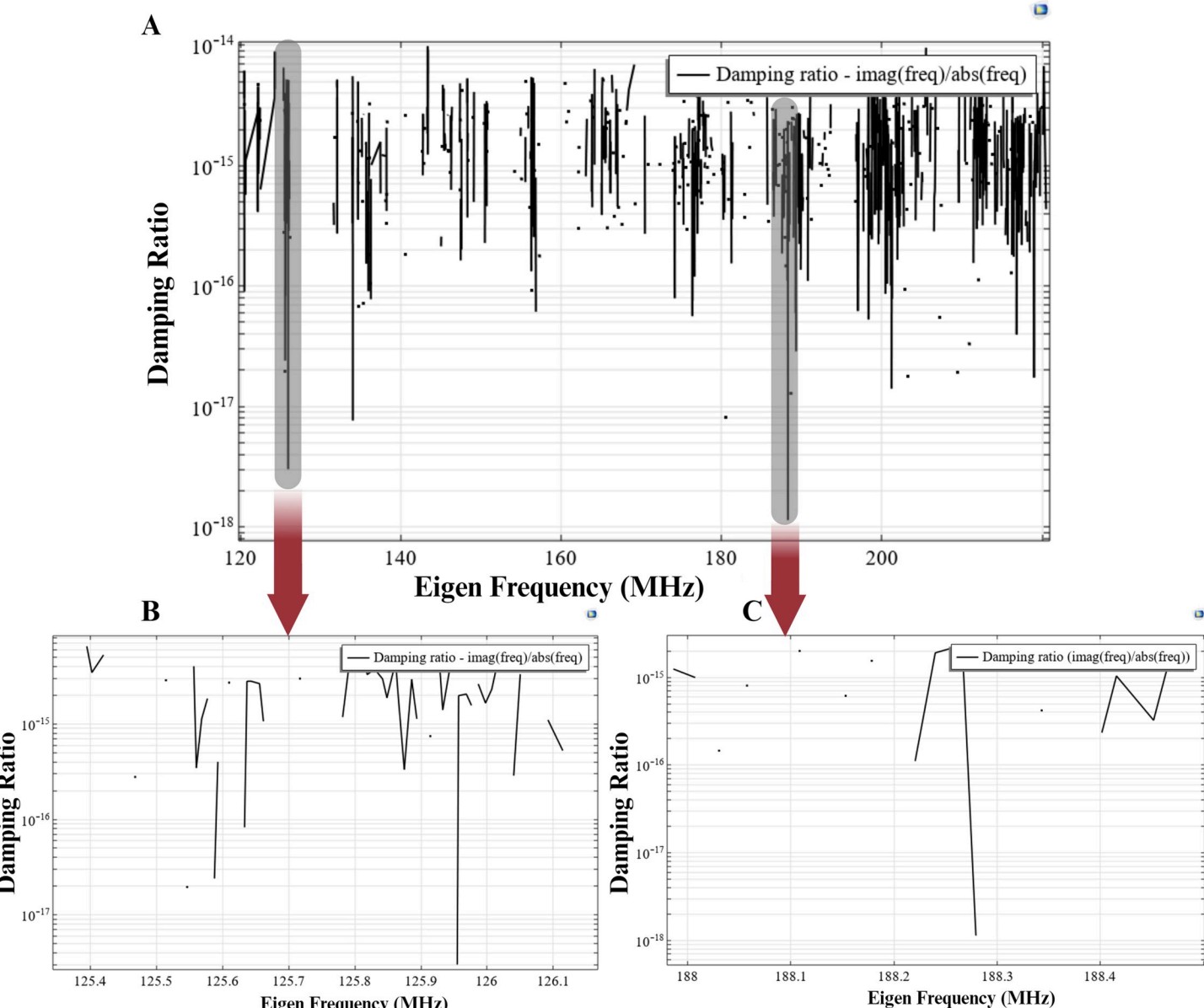

**Fig 5. A.)** Damping ratio for different Eigen frequencies (30–240 MHz) for the ME core shell, externally biased at 293 $mT$. **B and C.)** The damping ratio for the selected AC excitation frequencies of 126 MHz and 188 MHz is lower than $10^{-15}$.

demodulation effects, we conducted the simulations at these higher frequencies. We then analyzed frequency demodulation at the difference frequency of 62 MHz.

The elastic deformation at sampled frequencies within the 30–240 MHz range is illustrated in Fig 6. As demonstrated, increased frequency leads to greater deformation and a higher number of peaks, which can result in a relative increase in polarization in the subsequent piezoelectric shell.

### Frequency demodulation

We conducted a time-domain simulation on the magnetostrictive core, incorporating both the two AC excitation frequencies and the DC bias. Fig 7 shows the magnetic field induced within

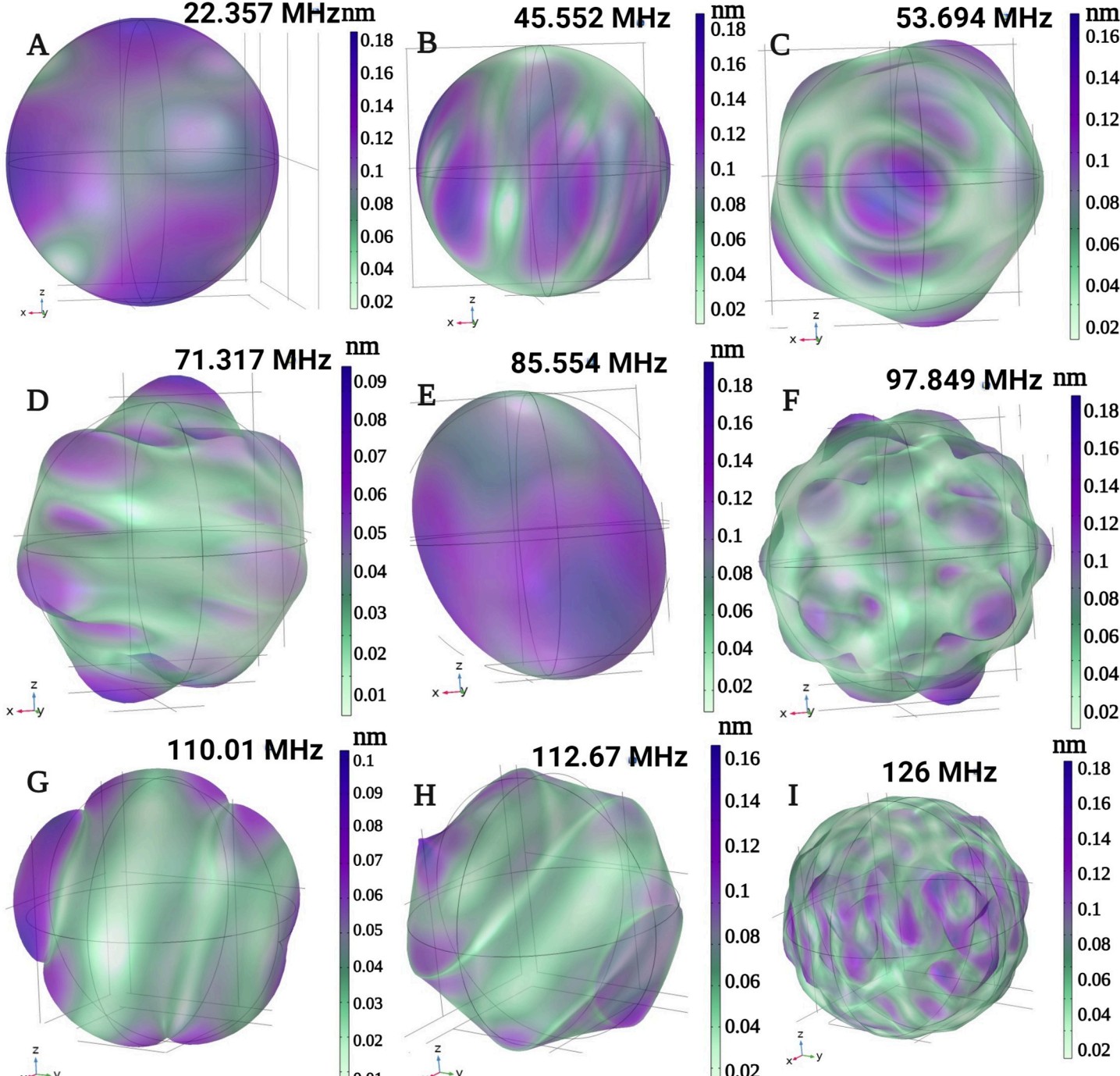

**Fig 6. Deformation of the ME core shell for different Eigen frequencies.** Irregular deformation at certain frequencies could generate the hypothesized electrical hotspots. Displacement in certain spots is comparatively higher than other spots, for certain Eigen modes, confirming our hypothesis.

the core at various time instances, as well as the resulting deformation of the core due to this induced magnetic field, as illustrated by the geometry of the sphere.

Magnetic Temporal Interference (MTI) can be generated by overlapping two different magnetic field frequencies, resulting in an amplitude-modulated (beating) waveform at the

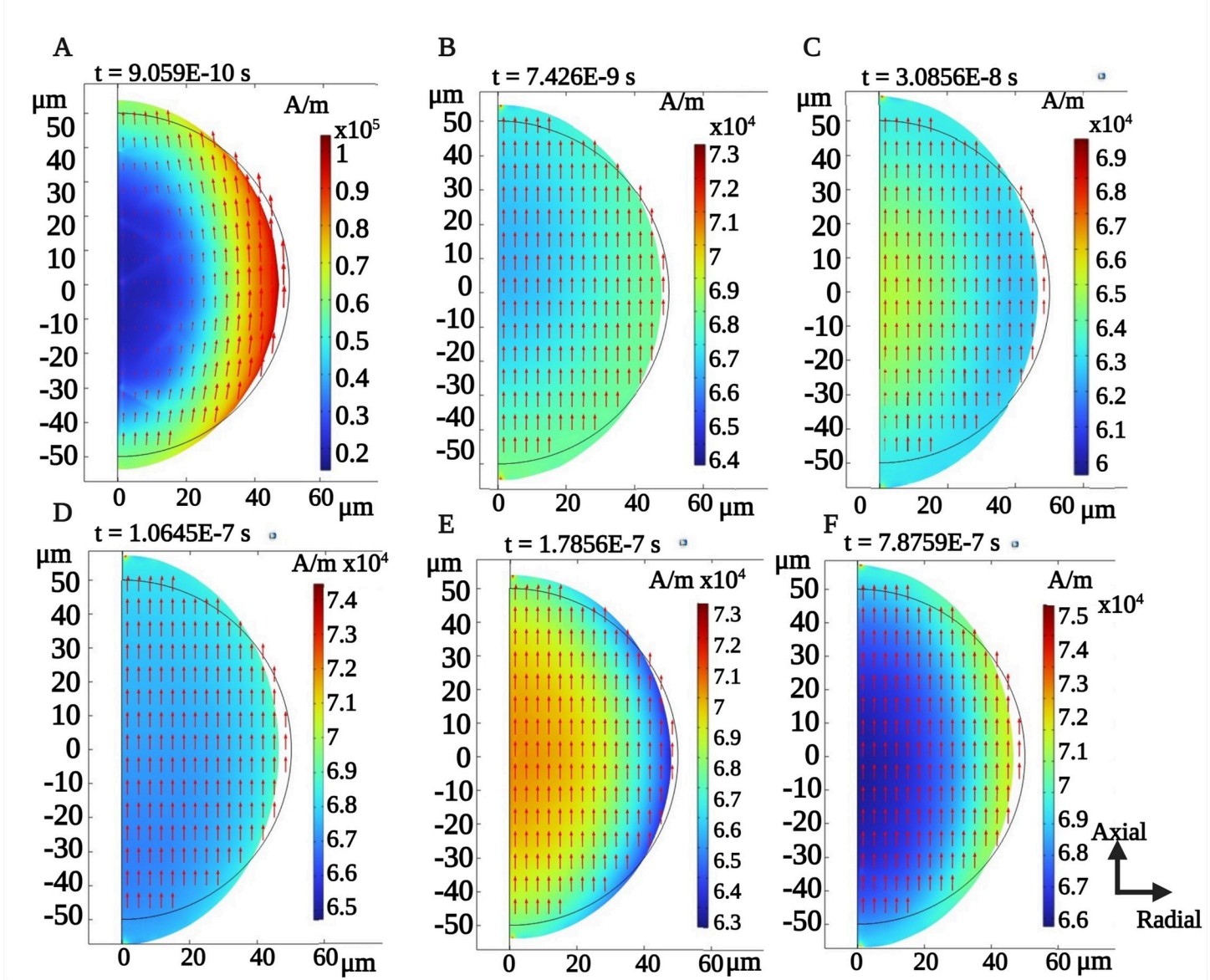

**Fig 7.** A 2D axis-symmetric simulation showing the normalized magnetic field induced on the magnetostrictive core (shown with color bar) for fixed DC bias of 293 mT (A-F). The deformation of the core due to the induced magnetic field as the movement in the circle boundary. The observed average magnetic field is appx. 66.7 kA/m.

difference frequency [69]. In our system, this MTI signal interacts with the magnetoelectric (ME) microdevice, producing a nonlinear multiplication of the applied frequencies. We demonstrate the impact of MTI through simulations in both an air medium and within the ME–MTI system. Fig 8 depicts the time-domain variations of the magnetic field within the magnetostrictive core (Fig 8A), as well as a spherical air medium in COMSOL (Fig 8C). The magnetostrictive core reveals waveform deformation, where the spectral density displays the core's nonlinear behavior in the magnetic field, resulting in the generation of a difference frequency at 62 MHz (Fig 8B), whereas the amplitude of the magnetic field in the air medium follows a beating envelope, with the power spectral density incorporating both

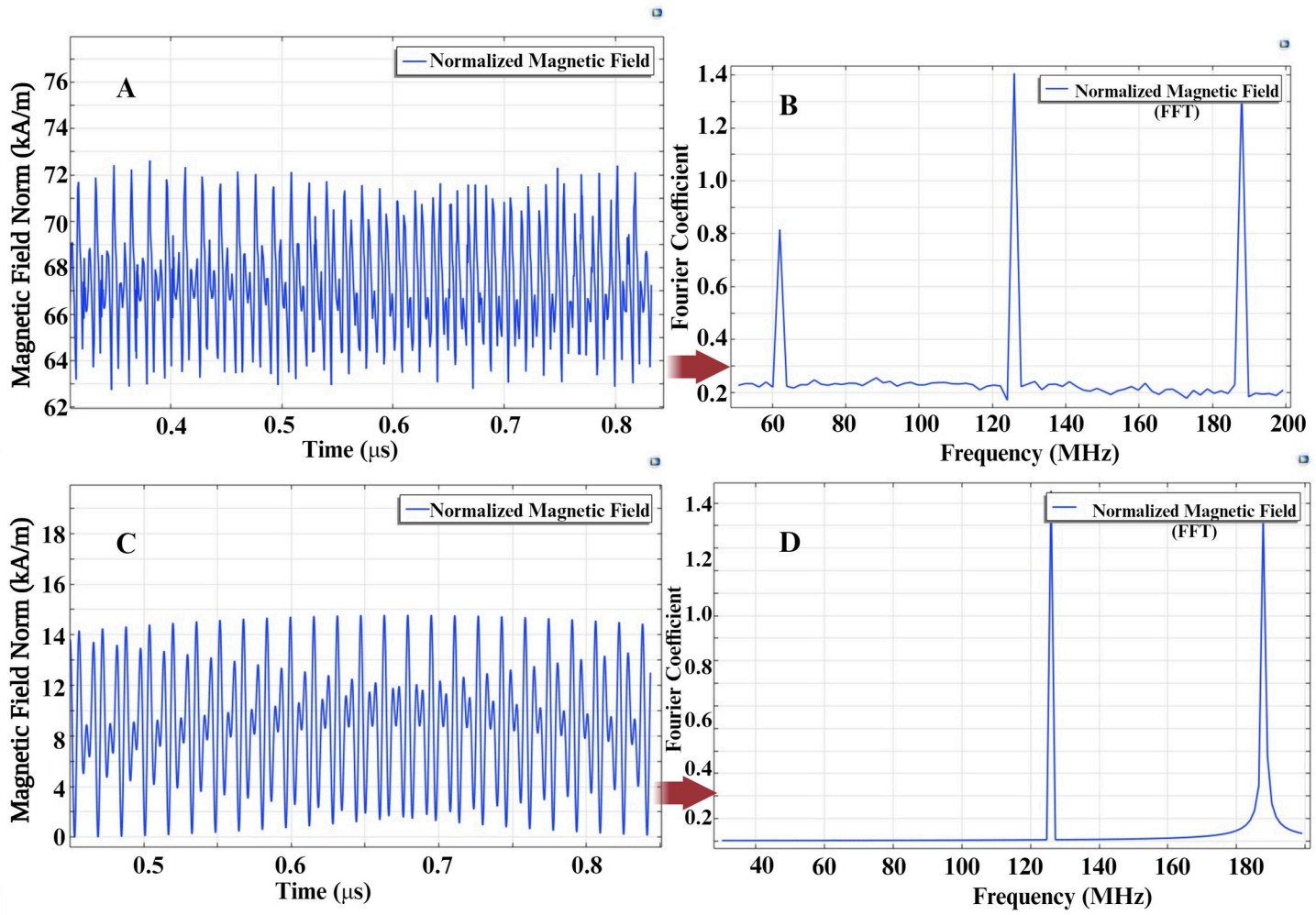

**Fig 8.** Normalized magnetic fields induced in **A)** the Magnetostrictive core and **B)** Power Spectral Density (FFT) on the core. Normalized magnetic fields induced in **C)** the Air and B) Power Spectral Density (FFT) on the air medium. The applied magnetic fields are at frequencies f1 (126 MHz) and f2 (188 MHz). The demodulated frequency component ($\Delta f = f2 - f1$) of 62 MHz is highlighted in B, demonstrating the ME-MTI system's unique frequency characteristics.

applied frequencies (Fig 8D). We have validated these findings using different frequency excitations, and the results consistently demonstrate the nonlinear demodulation effect.

## Electric field and ME coefficient

In this section, we explore the electric field distribution and the magnetoelectric coupling achieved through the addition of a piezoelectric shell. We examined the distribution of the induced electric field and polarization across this piezoelectric layer due to the applied magnetic field.

### Electric field distribution

An aluminum nitride (AlN) piezoelectric coating was integrated with the magnetostrictive core to transduce its mechanical deformations into stress within the piezoelectric shell. Computations were performed using COMSOL, incorporating coupled Multiphysics domains of solid mechanics, electrostatics, and magnetic fields in single time domain simulations. The

piezo shell's thickness was optimized to 37.5 $\mu m$, guided by the coupling coefficient data (Fig 11). Fig 9 illustrates the resulting electric field distribution on the piezo shell, generated by the applied magnetic field to the medium. We observed dynamic patterns in the electric field across the piezo shell, corresponding to variations in the core magnetic field. Due to the multi-modal nature of these magneto-mechanical deformations, we identified multiple intensified electric field distributions on the piezo shell, especially (see Fig 9). These irregular deformations, follows the same non-linear behavior which stem from the core material's non-linear magnetostrictive properties, led to a localized, non-linear electrical "hotspots" on the piezoelectric shell. Meaning that the electric field observations indicate the same frequency conversion observed in the magnetic field of the core device.

Moreover, Fig 10 showcases the time-dependent average and peak electric fields, measured on the surface of the piezo shell. The maximum surface electric field recorded ranged between 20 and 70 $mV/\mu m$. Additionally, at specific time points and locations, the electric field exceeded 100 $mV/\mu m$.

## Magnetoelectric coupling coefficient

The magnetoelectric (ME) coupling factor ($\alpha_{ME}$) serves as a crucial metric for understanding the non-linear interaction among the magnetic, elastic, and electric domains within the core-shell structure. This factor quantifies the transformation of induced magnetization into electrical polarization and is mathematically defined as:

$$\alpha_{ME} = \frac{dE}{dH} \left( \frac{V}{m} \cdot Oe \right), \tag{11}$$

where **E** and **H** represent the electrical and magnetic fields generated in the piezoelectric and magnetostrictive layers, respectively.

Based on our simulations, variations in the applied $H_{DC}$ and piezo shell thickness influenced the value of $\alpha_{ME}$. Fig 11A depicts how $\alpha_{ME}$ varies for shell thicknesses ranging between 35 and 45 $\mu m$, at the difference frequency of 62 MHz, and for a DC bias range of 0–1000 $mT$. Fig 11B zooms in on this relationship by fine-tuning the piezo thickness between 37 and 37.7 $\mu m$, in increments of 0.1 $\mu m$, to pinpoint an optimal coupling coefficient. A peak coupling factor of approximately $\alpha_{ME} = 550 \frac{V}{m}$. $Oe$ was observed at a shell thickness of approximately 37.5 $\mu m$, as shown in Fig 11B.

Interestingly, within the operational DC bias range of 200–350 $mT$ (indicated by the shaded region), $\alpha_{ME}$ peaks, suggesting minimal secondary losses such as thermal power loss. This peak value indicates that maximum conversion efficiency has been attained. In summary, our work successfully achieved a maximal magnetoelectric coupling, obtained by optimizing both the DC bias and piezoelectric shell thickness.

## ME core shell for bio-stimulation

In this section, we incorporate the core-shell Magneto-Electric (ME) microdevice into a biological environment populated with neurons to explore its capabilities for neural activation. We utilize the well-established Hodgkin–Huxley (HH) neuron model [70, 71] for our simulations, aiming to assess how the electrical potential difference generated on the surface of the piezoelectric shell, as detailed in previous sections, can trigger neural action potentials. It is crucial to highlight that we have adjusted the stimulation frequency to fall within a biologically relevant range of 35–380 Hz for inducing action potentials in neurons. This is different from the 62 MHz MTI frequency employed in our earlier COMSOL simulations, which was chosen

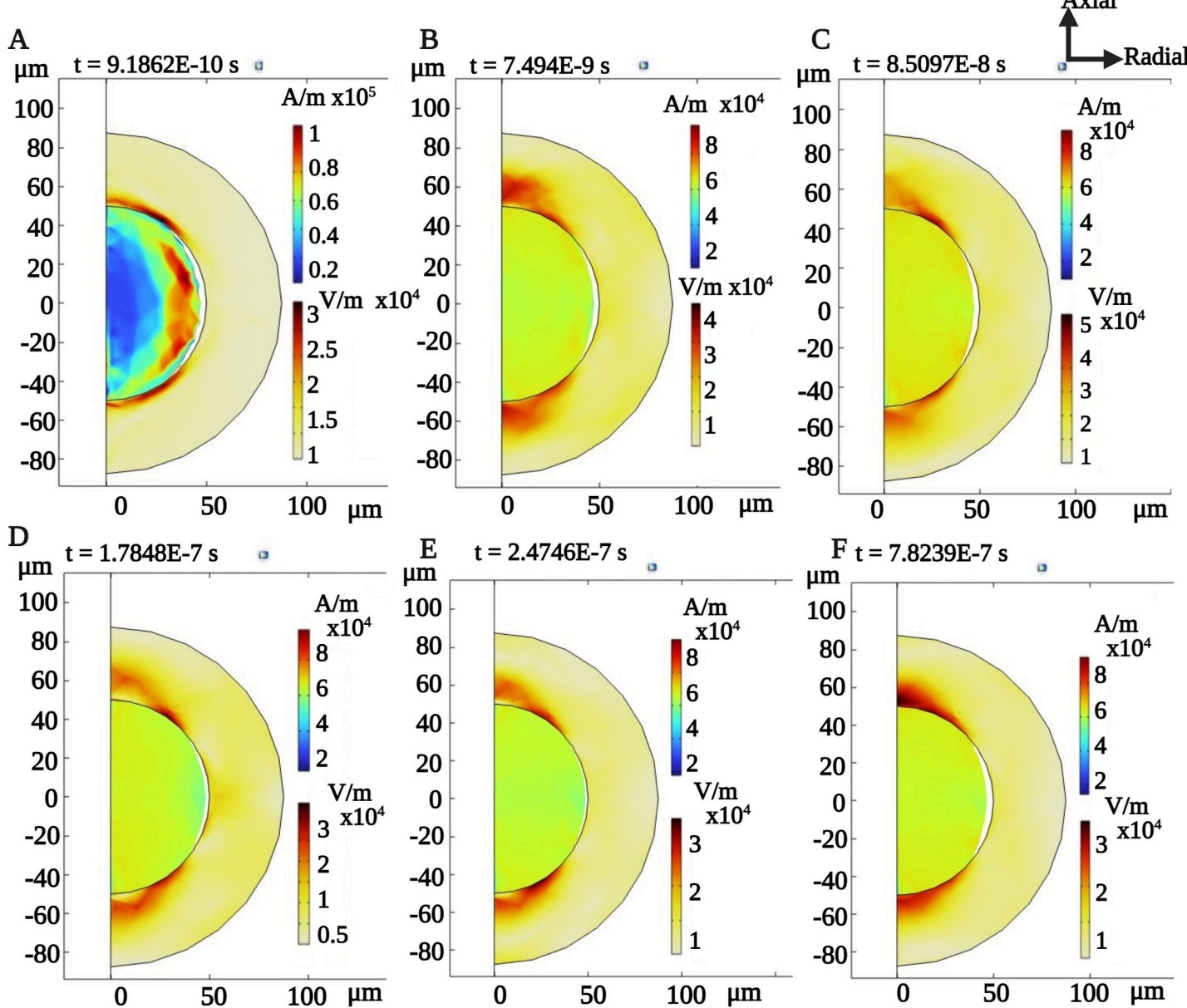

**Fig 9. Normalized magnetic field induced on the magnetostrictive core (top scale, in A/m) and the normalized electric field (bottom scale, in V/m) induced on the piezo shell (37.5 $\mu$m thickness) for an applied DC bias of 293 mT, at the MTI frequency of 62 MHz.** (A-F). The non-linear induced electric field is shown as the varying bright red spots on the piezoelectric shell, at varying time instances.

for computational efficiency and memory-saving purposes. To resolve this, we introduce a corresponding difference frequency for the external signals that are applied to the coils.

Fig 12 provides a graphical representation that visualizes the interaction between two arbitrary points on the core-shell device and a neuron's membrane surface. This figure also includes a generic circuit model to conceptualize the intricate bio-electro-mechanical interactions involved.

In HH model, the neuronal electrical activity is described through the membrane potential ($v_m$), which is dependent on voltage-gated potassium (K+) channels, voltage-gated sodium

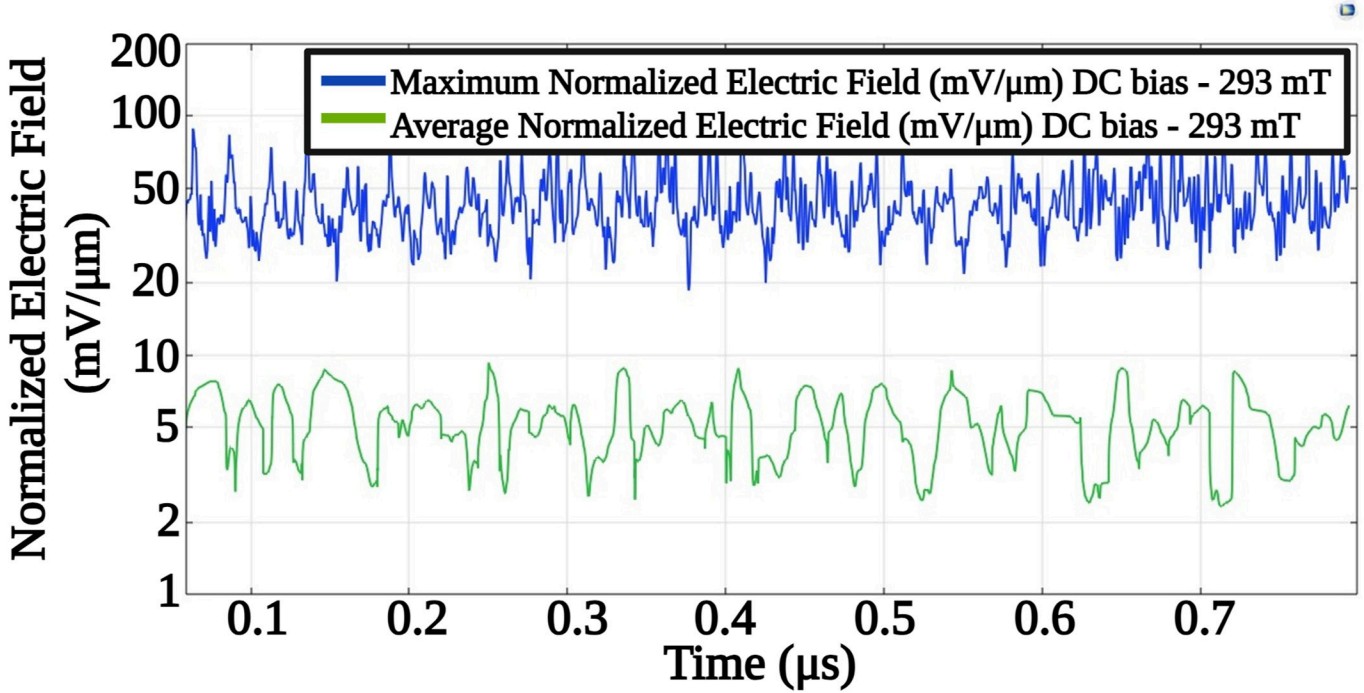

**Fig 10. Surface maximum and surface average electrical field values on the Piezo shell (37.5 $\mu m$ thickness), for an applied DC bias of 293 mT.**

(Na+) channels, a leak current, and an induced current ($i_{ind}$) [70]:

$$\frac{\mathrm{d}\nu_m}{\mathrm{d}t} = -\frac{1}{c_m}[g_K(\nu_m - V_K) + g_{Na}(\nu_m - V_{Na}) + g_L(\nu_m - V_L) - \underbrace{i_{ind}}_{\text{stimulation}}], \qquad (12)$$

where $C_m$ is specific membrane capacitance; $V_K$, $V_{Na}$, and $V_L$ are Nernst potentials for K
+ ions; Na+ and other ions are combined as "leak" channels; and $g_k$, $g_{Na}$, and $g_L$ are the corresponding membrane conductances. Voltage-gated conductances $g_k = \bar{g}_K m_K^4$ and $g_{Na} = \bar{g}_{Na} m_{Na}^3 h_{Na}$ change with time during an action potential. $m_K^4$ and $m_{Na}^3 h_{Na}$ represent the opening probabilities for K+ and Na+ channels, respectively. The gating variables $m_K$, $m_{Na}$, and $h_{Na}$ and the relevant parameters are define here [70]. For the reproducibility of the results, we have also provided the Github link for the MATLAB file used to create the spike voltages [72].

In Fig 13, we display the electric potential difference between two random points on the piezoelectric shell surface, with peak-to-peak potentials ranging between approximately 250–400 $mV$. Given that the MTI frequency resides in the high-frequency (HF) spectrum (64 MHz), the pulse width for a single signal train is around 200 $ns$. This duration is insufficient for effective electrical neural stimulation, however we time scale signal to fall into the neural stimulation frequency range.

To calculate the injected current at the biological environment, and use the HH model, we calculated the static resistance between two fixed points on the cell surface by using numerical full wave EM simulations using CST Microwave Studio ® Finite Element Method (FEM) tool. The ohmic impedance at the low frequency range for two spaced electrodes of area 400nm$^2$ and the distance of 3 $\mu m$ is in the range 0.45–1 MΩ. This information is crucial for optimizing

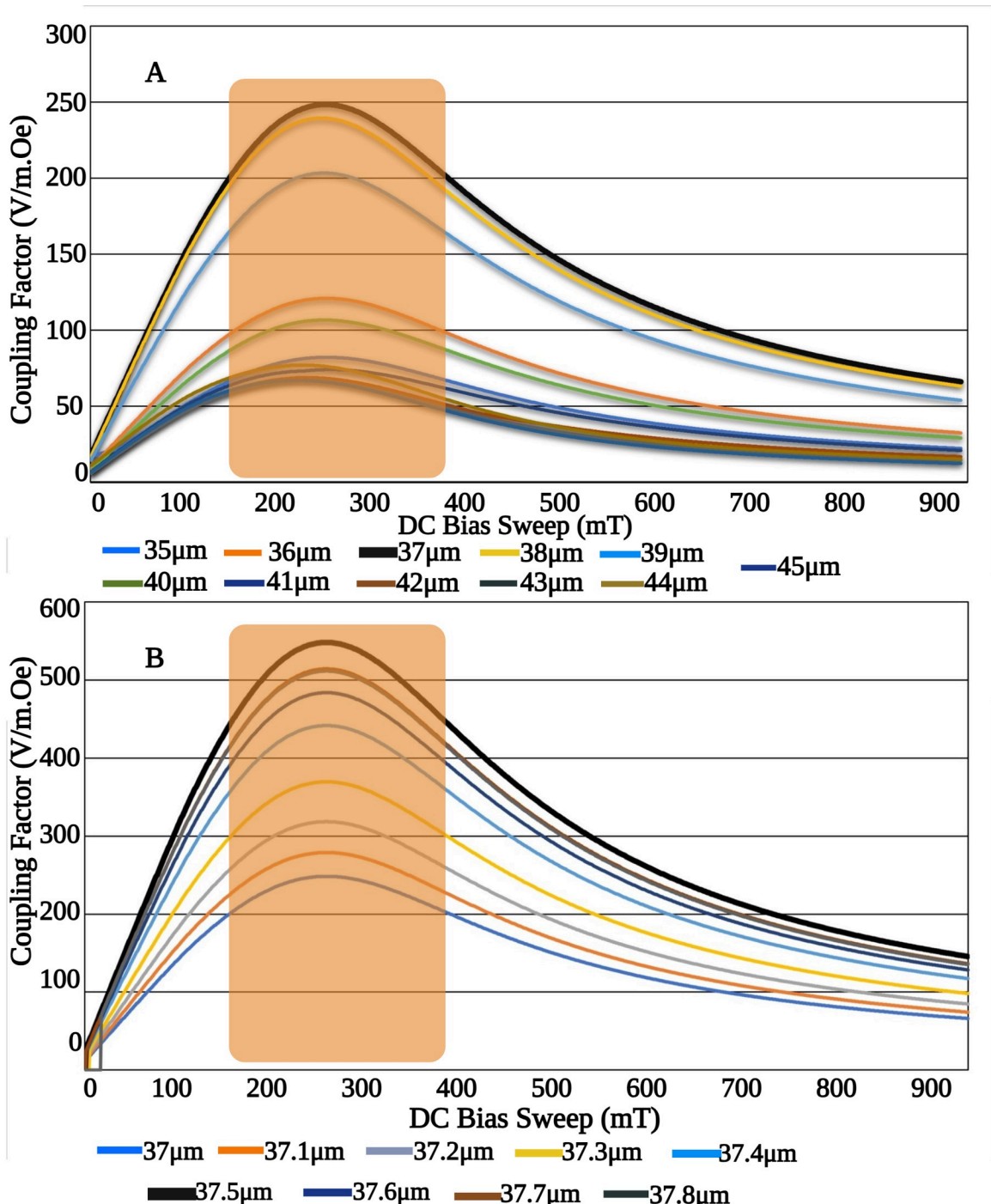

**Fig 11. A)** ME coupling factor analysis for 35–45 $\mu$m (top) and **B)** step reduced analysis (bottom). The optimum Piezo shell thickness is determined to be 37.5 $\mu$m at which maximum $\alpha_{ME}$ is obtained. The shaded region shows the area within which $\alpha_{ME}$ is maximum.

the interaction between the core-shell ME microdevice and the biological medium for potential neural stimulation applications.

The current signal, $i_{ind}$, comprises various frequency components, which include the two initially applied RF frequencies, the temporal interference frequency, as well as the harmonics

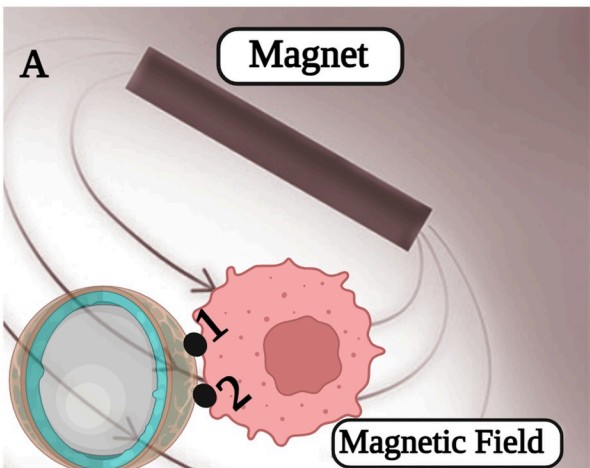
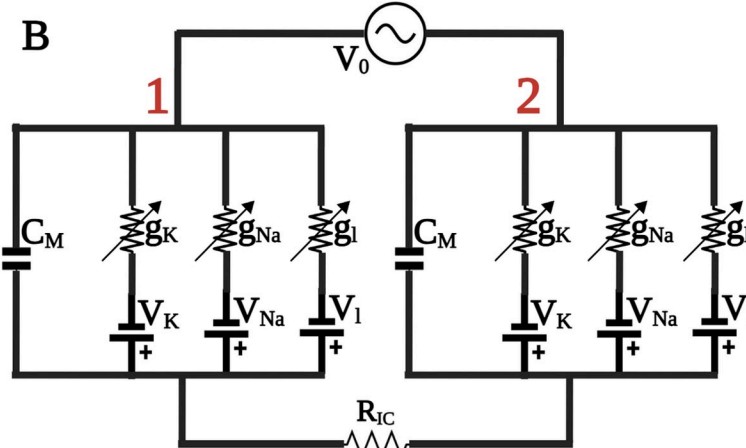

**Fig 12.** **(A)** Schematic of the core shell interaction with a cell membrane and **(B)** the equivalent circuit representing the cell. V0 is the voltage difference between two random points on the core shell interacting with the cell. RIC is the intra-cellular resistance inside the cell.

generated by the device's non-linearity. While the original applied RF frequencies are not capable of cellular stimulation, the difference frequency serves as the signal responsible for stimulation. To isolate this component, a low-pass filter with a 100 MHz cut-off is applied to remove the RF signal from the stimulation pattern.

To analyze the impact of this filtered signal on neural spike generation, the time scale was adjusted by a factor of $10^6$. This adjustment compensates for the high-frequency difference (62 MHz) selected in the simulation, effectively mapping the MTI frequency to 62 Hz for biological relevance. When this filtered and time-scaled current signal is input into a HH model, it becomes evident that the signal's amplitude substantially exceeds the necessary stimulation threshold. To address this, the amplitude is scaled down within a range of 200–5000, and the resulting number of membrane potential spikes (shown in Fig 14) is then counted.

Fig 15 illustrates the number of generated spike potentials across various amplitude scaling factors. It's worth noting that this amplitude scaling is functionally equivalent to either reducing the current in the external coil relative to the defined distance between the coils and the Magneto-Electric (ME) device or increasing the separation distance between the external applicator and the implanted ME device while maintaining the same current. As demonstrated in Fig 15, a scaling factor of 200 resulted in a high number of generated spikes, but also raised the risk of hyper-polarization or premature depolarization. As we increased the scaling factor, we observed a plateau phase featuring stable and temporally consistent spike generation, where the peak amplitude of $i_{ind}$ reached 15 $\mu A/cm^2$. Additionally, when $i_{ind}$ dropped below 8 $\mu A/cm^2$, spike generation became increasingly irregular, and no spikes were generated when $i_{ind}$ was less than 3 $\mu A/cm^2$. Consequently, we conclude that for effective localized cellular or tissue stimulation via the ME core-shell device, the external parameters should be tuned to produce an $i_{ind}$ value falling within the shaded region shown in the figure. We also estimate that an exposure time of approximately 10 $ms$ is sufficient to elicit a single spike.

## Discussion

The Magnetoelectric (ME) effect, with its promising applications in medical sensing and stimulation, is gaining momentum in research. While most ME research has been devoted to characterizing and quantifying material properties, the potential of non-linear magnetostriction in

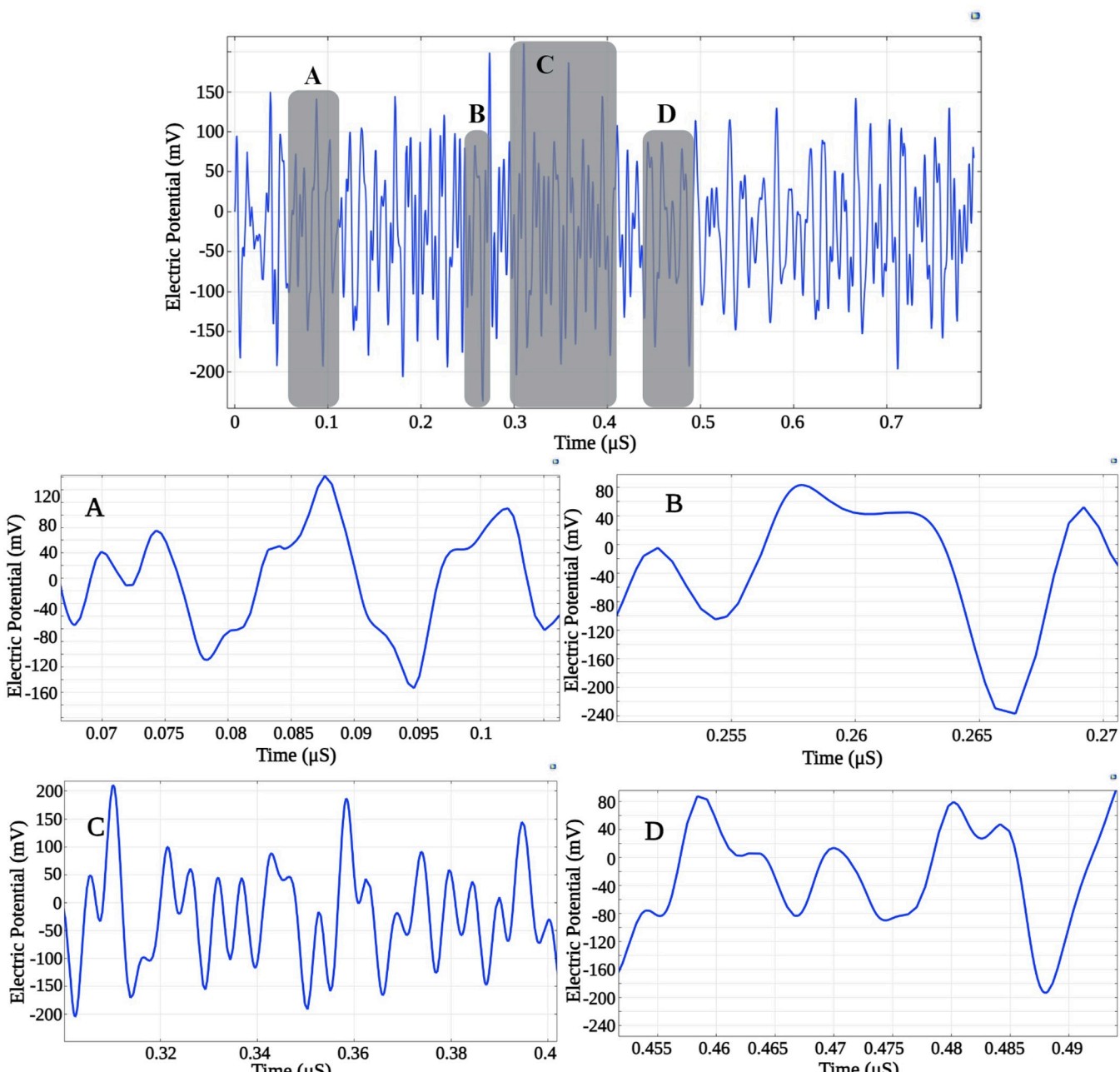

**Fig 13. Time variation of electric potential difference between two random points on the piezo shell surface.** The shaded areas are expanded and shown related to regions A—D.

ME cores for frequency demodulators remains relatively untapped. In this study, we introduce a novel battery-free ME core-shell structure. This design enhances bio-stimulation capabilities and eliminates the need for additional circuitry for frequency demodulation. We highlight that the non-linear traits of ME core shells can facilitate demodulation, allowing for the extraction of low-frequency components from AC excitations. Our results provide a foundation for understanding ME core shells in resonant operation and their potential applications in medical patch sensing and implantable devices.

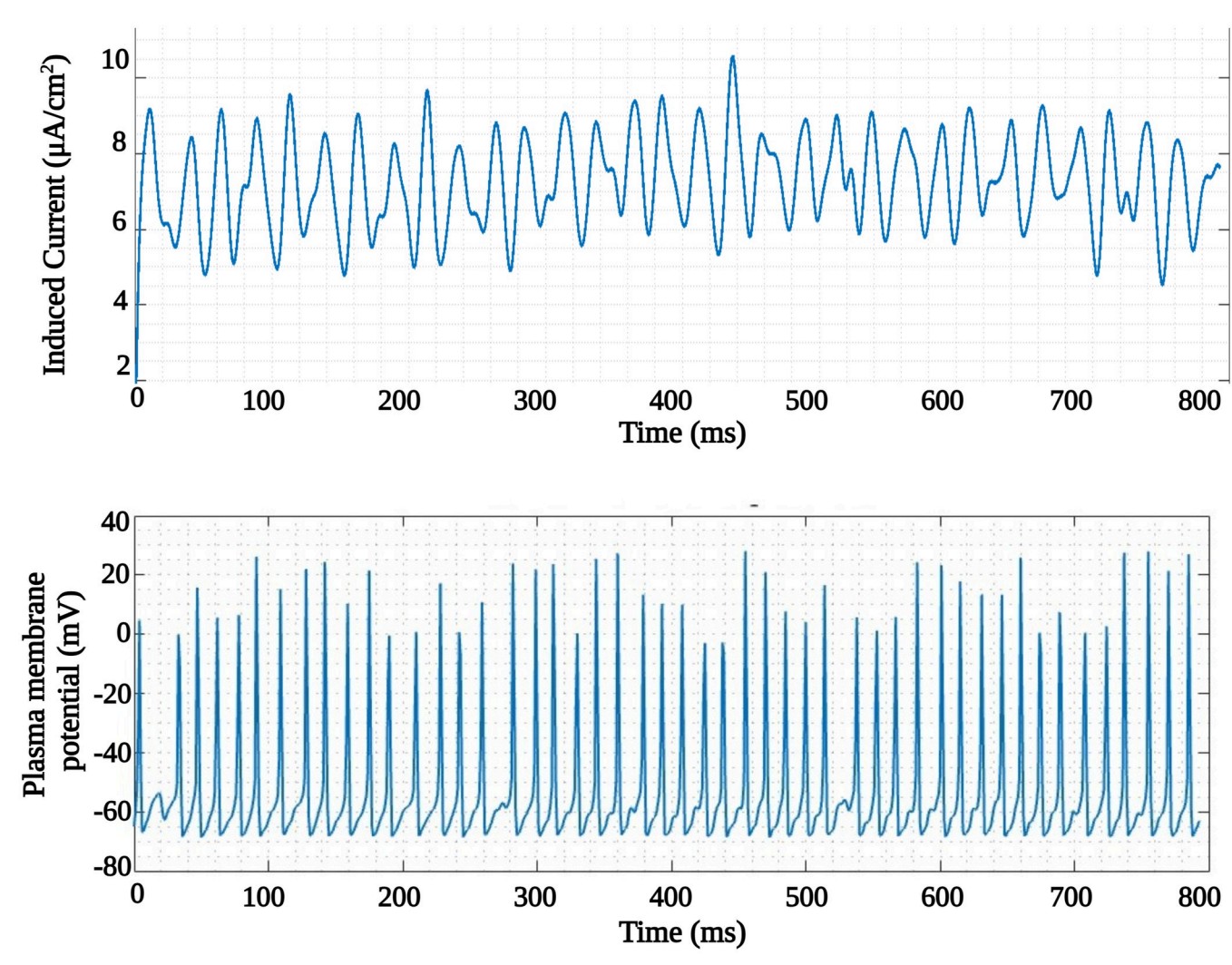

**Fig 14. Induced current and the evoked plasma membrane potential using amplitude scaling factor.** For the applied induced current spike, a neural spike was generated, conforming to the voltage levels of the repolarization (-70 $mV$) and depolarization (30–40 $mV$).

To exploit this potential of ME core shells, multiple control parameters must be taken to consideration, which include, the core radius, magnetostriction of the core material, piezo shell thickness, placement of the external AC excitation coils and the DC bias value. Moreover, the operation of the ME in its Eigen frequencies, amplifies the magnetostriction induced deformation of the core and subsequently increases the electrical field distribution on the piezo layer.

In our fundamental simulation, two coils were placed symmetrical to the MetGlas core and excited with two different AC magnetic fields. The DC magnetic bias was fixed to a constant value, where maximum magnetostriction was observed. During the perturbation process, the core deforms as an intermediate result of the externally applied magnetic fields. The random compression and expansion in the axial and radial axes are a result of the placement of the AC excitation coils and the applied DC magnetic bias. Controlled change in the properties of the

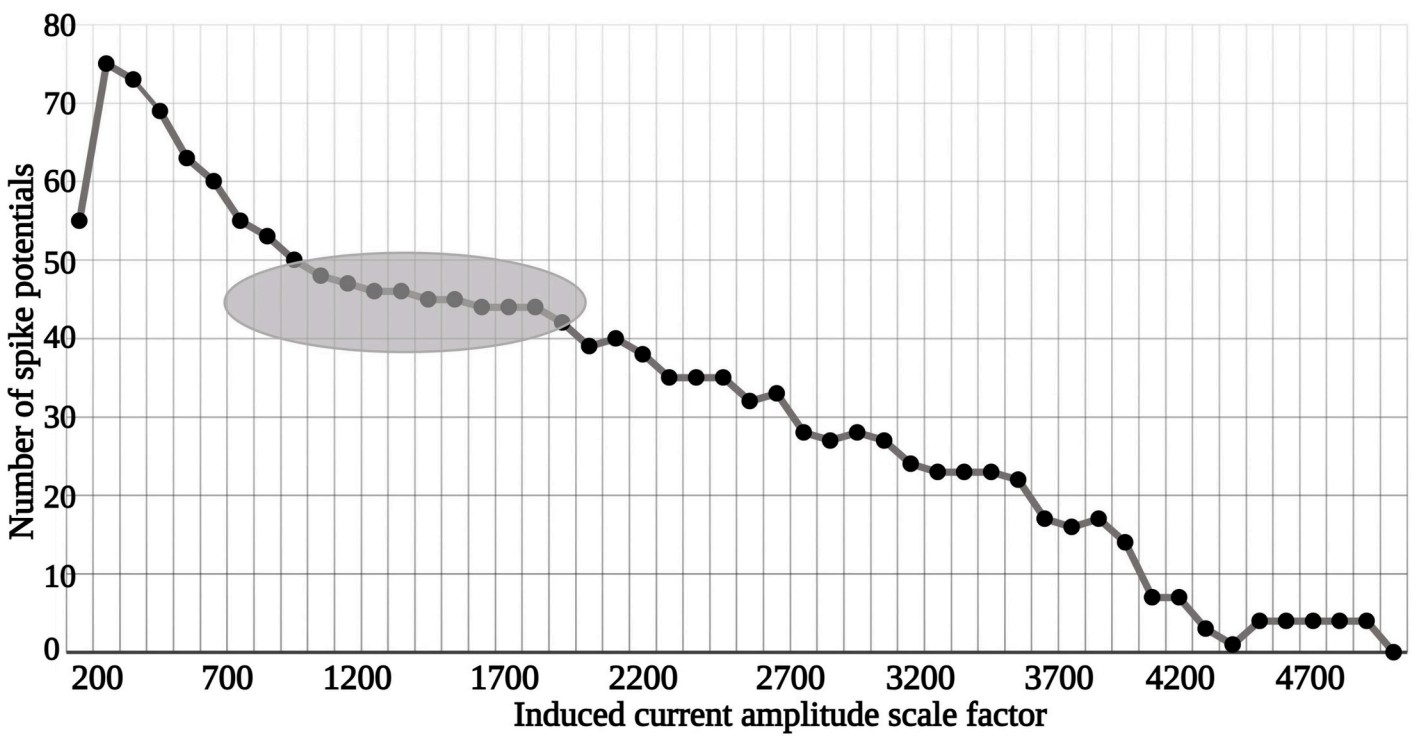

**Fig 15. This graph displays the number of generated spike potentials as a function of the scaled amplitude values of $i_{ind}$.** The shaded region highlights the plateau where spike generation stabilizes.

external DC and AC fields, could tune the deformation in the magnetostrictive core. In the subsequent simulations, we added the piezo shell layer and optimized its thickness for maximum coupling. The strain generated on the magnetostrictive core is transferred to the piezo shell as an electrical field distribution. Fig 11 shows the variation in the ME coupling coefficient for different piezo thickness. The Eigen frequency analysis was carried out to select the two AC excitation frequencies with minimum loss. At resonance, the deformation of the core is maximum, resulting in maximum electrical field on the piezo shell. To justify the selected AC excitation frequencies, we analyzed the damping factor for the available mechanical resonances in a range of selected frequencies (from 30–240 MHz). Two frequencies with damping ratios lower than $10^{-15}$, and with a substantial bandwidth difference, were selected. The mechanical resonances were non-uniform in many modes, which is also an advantage, so that, with precise placement of coils, selection of frequencies, the resultant electrical field distribution on the piezo shell could be controlled on specific areas on the shell. These predefined areas on the shell could be in contact with the cell or tissue layers (as depicted in Fig 12A). It should be noted that resonance frequencies vary for different dimensions of the core shell.

ME devices can be exploited for their multifunctional use through their non-linear eigenmodes. An interesting application could involve targeted drug delivery to neural tissue, where non-resonant frequency excitations could loosen the bonds between the drug and the ME particle [61]. Subsequent excitation of the ME device at its resonance could trigger electrical pulses on the piezo shell for bio-stimulation.

ME nanoparticles were experimentally tested in low-frequency non-resonant modes [56], where the use of such particles as a coagulated entity was experimentally tested. While the use of ME core shells in the nanoscale is required for cellular uptake and to cross the Blood-Brain

Barrier, the controllability of single entities in a coagulation is complex. Our design suggests the use of micro core shells that adhere to the mechanical resonant modes, enabling future fabrication experiments to fabricate subsets of core shells with slightly different dimensions. This allows for intricate tuning of the external magnetic fields to control each of the subset which are identified through their unique resonant modes.

Additionally, the modeling of single ME core-shell nanoparticle was recently published by Fiocchi et al. in [45]. They achieved a coupling of 0.28 V/cm·Oe with their ME core-shell nanoparticles. In contrast, our design achieved a coupling of approximately 5.5 V/cm·Oe. This increase is primarily attributed to the larger dimensions of our core shell and the operation of the ME device in mechanical resonance modes. It is important to note that reducing the size of our core shells to the nanoscale increases mechanical resonance, indirectly increasing the complexity of the external coils and the required circuitry to process GHz/THz frequency inputs. We utilize the mechanical resonance modes of our designed core shells, which are in the order of the 100 MHz range. Therefore, we limit our size to the microscale to remain within the MHz frequency range. We fixed the MetGlas core to be $50\mu m$ in radius and then characterized the magnetostrictive core for its maximum magnetostriction.

To confirm the validity of our simulation models, we first simulated an existing experimental study of a ME structure. Joy et al. analyzed a miniature ME antenna as a planar structure [68], and their first step was to calculate the optimum DC field at which the structure needed to be biased. They conducted experimental verification and obtained a DC bias value. We replicated the same structure in COMSOL and observed that we also obtained the same DC bias field calculated through their experiments. This confirmed the validity and boundary conditions of our modeling for further simulations.

## Conclusion

This study provides a proof-of-concept and analytical methodology, demonstrating the use of a Magnetic Temporal Interference for frequency demodulation through non-linear ME core shells to stimulate biological cells and tissues. Our central discovery is the ability of these remotely powered, resonant ME core shells to produce electric fields potent enough to stimulate cells and tissues effectively. Capitalizing on the irregular deformation and non-linearity of magnetostriction, it is possible to precisely localize the electric field on the piezoelectric shell. This precise localization is achieved through strategic adjustments in geometry, magnetic bias control, and frequency selection. By fine tuning external excitation frequencies, we can induce optimal temporal interference low-frequency magnetic fields on the magnetostrictive core, thereby generating electric fields on the piezo shell, tailored for cellular stimulation.

Our proposed method, termed ME-MTI, offers notable advantages over conventional Magnetic Temporal Interference. Notably, it provides enhanced spatial resolution and energy efficiency, leading to a reduction in the size and complexity of external units. The capability of ME core shell microdevices to produce low-frequency generation and demodulation responses means that cells and tissues can effectively filter out high-frequency fields, relying solely on MTI fields for stimulation. This research underscores the promise of ME core shells in remote neural stimulation and brain control applications, especially pertinent for deep brain stimulation or for inhibiting specific sensations in the brain cortex. However, further validation is essential. Future investigations should focus on analyzing the thermal effects of the implantable microdevice, evaluating the Specific Absorption Rate (SAR) of the external coils, and exploring the broader potential of ME-MTI core shells in the realm of biomedical applications.

## Acknowledgments

All figures were either created or edited with BioRender.com.

## Author Contributions

**Conceptualization:** Ali Khaleghi.

**Formal analysis:** Ram Prasadh Narayanan.

**Funding acquisition:** Ilangko Balasingham.

**Investigation:** Ram Prasadh Narayanan, Mladen Veletić.

**Methodology:** Ram Prasadh Narayanan, Ali Khaleghi.

**Project administration:** Ilangko Balasingham.

**Resources:** Ilangko Balasingham.

**Supervision:** Ali Khaleghi, Ilangko Balasingham.

**Validation:** Ram Prasadh Narayanan, Mladen Veletić.

**Writing – original draft:** Ram Prasadh Narayanan.

**Writing – review & editing:** Ram Prasadh Narayanan, Ali Khaleghi, Mladen Veletić, Ilangko Balasingham.

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
