## [Decision Letter · Decision Letter 0]

14 Nov 2023

PONE-D-23-33833Multiphysics Simulation of Magnetoelectric Microdevices for Wireless Cellular Stimulation Therapy via Magnetic Temporal InterferencePLOS ONE

Dear Dr. Narayanan,

Thank you for submitting your manuscript to PLOS ONE. After careful consideration, we feel that it has merit but does not fully meet PLOS ONE’s publication criteria as it currently stands. Therefore, we invite you to submit a revised version of the manuscript that addresses the points raised during the review process.

We look forward to receiving your revised manuscript.

Kind regards,

Yuan-Fong Chou Chau

Academic Editor

PLOS ONE

Journal Requirements:

2. Thank you for stating the following financial disclosure: "Brain-Connected inteRfAce TO machineS (B-CRATOS) under grant #965044, Horizon 2020 FET-OPEN. 

Research Council of Norway under the grant #287112 (CIRCLE – Communication Theoretical Foundation of Wireless Nanonetworks)

ERCIM ‘Alain Bensoussan’ Fellowship Programme".

3. Please expand the acronym “ERCIM” (as indicated in your financial disclosure) so that it states the name of your funders in full.

4. Thank you for stating the following in the Acknowledgments Section of your manuscript: "The work has been supported by the project Brain-Connected inteRfAce TO machineS (B-CRATOS) under grant #965044, Horizon 2020 FET-OPEN. This work was funded in part by the Research Council of Norway under the grant #287112 (CIRCLE – Communication Theoretical Foundation of Wireless Nanonetworks). This work was carried out during the tenure of the ERCIM ‘Alain Bensoussan’ Fellowship Programme awarded to the first author. All figures were either created or edited with BioRender.com. "

Please remove any funding-related text from the manuscript and let us know how you would like to update your Funding Statement. Currently, your Funding Statement reads as follows: "Brain-Connected inteRfAce TO machineS (B-CRATOS) under grant #965044, Horizon 2020 FET-OPEN. 

Research Council of Norway under the grant #287112 (CIRCLE – Communication Theoretical Foundation of Wireless Nanonetworks)

ERCIM ‘Alain Bensoussan’ Fellowship Programme"

Reviewers' comments:

Reviewer's Responses to Questions

**Comments to the Author**

1. Is the manuscript technically sound, and do the data support the conclusions?

Reviewer #1: No

Reviewer #2: Partly

Reviewer #3: Yes

2. Has the statistical analysis been performed appropriately and rigorously? 

Reviewer #1: Yes

Reviewer #2: Yes

Reviewer #3: Yes

3. Have the authors made all data underlying the findings in their manuscript fully available?

Reviewer #1: Yes

Reviewer #2: Yes

Reviewer #3: Yes

4. Is the manuscript presented in an intelligible fashion and written in standard English?

Reviewer #1: Yes

Reviewer #2: Yes

Reviewer #3: Yes

5. Review Comments to the Author

Reviewer #1: The manuscript presents an innovative approach by combining MTI with ME particles for neural stimulation, a synergy not previously explored. However, the paper currently only offers simulation results. The structural organization of the manuscript needs significant enhancement. The text requires thorough revision for clarity and flow to reach a publishable standard.

Major Points for Revision:

1. The rationale for employing MTI signals for neural stimulation should be clarified early in the manuscript. Comparative simulations with both MTI and single-frequency magnetic fields could illustrate the advantage of MTI in generating larger electric fields around the ME particles.

2. The literature review in the introduction does not seem exhaustive.

3. Specific comments on the manuscript:

• Line 64: Reference 16 appears misplaced as it does not pertain to implantable micro-coils as suggested.

• Line 72: The claim of novelty in the magnetoelectric Multiphysics behavior should be substantiated. While similar works are cited later (e.g., Ref. 210), these should be integrated into the discussion at this point to establish the context of novelty effectively.

• The practicability of the proposed ME particles is questionable given the requirement for two large coils on either side. A discussion on how this setup could be realistically implemented in a clinical setting is necessary.

• Incorporating an analysis of thermal effects within the current study would greatly enhance the research's value, especially given the significant concern of tissue temperature increases due to a large (100μm) ME particle.

Minor Points for Revision:

1. The term 'microdevices' used to describe the ME particles could be misleading since the technology lacks electronic components and the broader scientific community typically refers to such entities as micro/nanoparticles.

2. Specific comments on the manuscript:

• Line 60: The manuscript could acknowledge the existence of novel TMS systems capable of deep tissue layer stimulation.

• Line 221: The assertion made here requires substantiation with a relevant reference.

• Line 504: The term "conventional magnetic temporal interface" seems to confuse "interface" with "interference." Moreover, a definition or citation for what constitutes conventional MTI is necessary for clarity.

3. In Figure 3, it would be beneficial to specify the distance between the coil and the ME particle.

Reviewer #2: Authors have proposed wireless stimulation of neurons with ME heterostructures. Although the principle of ME coupling is simple, the deformation of magnetostrivtive core drive the piezeoelectric shell to supply electric response. I have a lot of concerns on the possibility of this concept for real applicaton:

1. We need ordered electric signals of ME nanoparticles to stimulate certain neurons. However, the polarization of the piezoelectric response depends on the poling directions (self-poled or externally poled). It means that the particle's electric response could cancel each other if they are in opposite poling conditions. So the key problem is how many particles is needed for stimulation of one neuron. If only one ME particle is needed, then it maybe useful. If more than one particles is needed, the accumulation effect of all particles should be carefully calculated and simulated.

2. I also worry about the safety of this therapy. Particles, although small, usually cause unpredictable and undesirable problems for biological tissues. Many of them could result in cancer. How to solve this problem in this therapy.

Reviewer #3: The authors presented a ME microdevice for wireless cellular stimulation therapy. This work is interesting, and the manuscript has shown improvement efforts. However, to enhance the quality of the study, the following minor questions should be addressed before it is acceptable for publication:

1. When applying AC magnetic field stimulation, it is recommended to address electromagnetic radiation safety thresholds. Authors also need to consider the attenuation of high-frequency AC magnetic fields within human tissue.

2. The polarization direction of the piezoelectric crystal should be clearly indicated. It would be beneficial to discuss the impact of magnetic field stimulation from different directions on the output electric field.

3. The material properties of MetGlas seem to be incomplete, particularly regarding the initial magnetic permeability.

6. PLOS authors have the option to publish the peer review history of their article (what does this mean?). If published, this will include your full peer review and any attached files.

Reviewer #1: No

Reviewer #2: **Yes: **Zhiguang Wang

Reviewer #3: **Yes: **Shuxiang Dong

---

## [Author Response · Author response to Decision Letter 0]

8 Dec 2023

Dear Reviewers, 

Once again, thanks for your time and the required reviews. It was helpful for us to understand expertise views. We have addressed your reviews in the manuscript. The support letter attached with this submission details our answers for your questions. For your further perusal. 

Thanks

Authors.

---

## [Decision Letter · Decision Letter 1]

29 Dec 2023

Multiphysics simulation of magnetoelectric micro core-shells for wireless cellular stimulation therapy via magnetic temporal interference

PONE-D-23-33833R1

Dear Dr. Narayanan,

We’re pleased to inform you that your manuscript has been judged scientifically suitable for publication and will be formally accepted for publication once it meets all outstanding technical requirements.

Kind regards,

Yuan-Fong Chou Chau

Academic Editor

PLOS ONE

Additional Editor Comments (optional):

Reviewers' comments:

Reviewer's Responses to Questions

**Comments to the Author**

1. If the authors have adequately addressed your comments raised in a previous round of review and you feel that this manuscript is now acceptable for publication, you may indicate that here to bypass the “Comments to the Author” section, enter your conflict of interest statement in the “Confidential to Editor” section, and submit your "Accept" recommendation.

Reviewer #1: All comments have been addressed

Reviewer #2: All comments have been addressed

2. Is the manuscript technically sound, and do the data support the conclusions?

Reviewer #1: Yes

Reviewer #2: Yes

3. Has the statistical analysis been performed appropriately and rigorously? 

Reviewer #1: Yes

Reviewer #2: Yes

4. Have the authors made all data underlying the findings in their manuscript fully available?

Reviewer #1: Yes

Reviewer #2: Yes

5. Is the manuscript presented in an intelligible fashion and written in standard English?

Reviewer #1: Yes

Reviewer #2: Yes

6. Review Comments to the Author

Reviewer #1: (No Response)

Reviewer #2: Authors have answered all my questions.Authors have answered all my questions.Authors have answered all my questions.Authors have answered all my questions.

7. PLOS authors have the option to publish the peer review history of their article (what does this mean?). If published, this will include your full peer review and any attached files.

Reviewer #1: No

Reviewer #2: **Yes: **Zhiguang Wang

---

## [Editor Report · Acceptance letter]

17 Jan 2024

PONE-D-23-33833R1 

PLOS ONE

Dear Dr. Narayanan, 

I'm pleased to inform you that your manuscript has been deemed suitable for publication in PLOS ONE. Congratulations! Your manuscript is now being handed over to our production team.

Kind regards, 

on behalf of

Dr. Yuan-Fong Chou Chau 

Academic Editor

PLOS ONE